# Novel Variants of SCC*mec* Type IX Identified in Clonal Complex 398 Livestock-Associated Methicillin-Resistant *Staphylococcus aureus* from Pork Production Systems in Korea

**DOI:** 10.3390/antibiotics14030217

**Published:** 2025-02-21

**Authors:** Gi Yong Lee, Soo In Lee, Hoon Je Seong, Soo-Jin Yang

**Affiliations:** 1Department of Veterinary Microbiology, College of Veterinary Medicine and Research Institute for Veterinary Science, Seoul National University, Seoul 08826, Republic of Korea; dominic3809@naver.com (G.Y.L.); dltndls1101@naver.com (S.I.L.); 2Korean Medicine Data Division, Korea Institute of Oriental Medicine, Daejeon 1672, Republic of Korea; hoonjeseong@gmail.com

**Keywords:** *S. aureus*, CC398 MRSA, SCC*mec* type IX, pork production systems

## Abstract

**Background/Objectives**: The occurrence of novel variants of staphylococcal cassette chromosome *mec* (SCC*mec*) in livestock-associated methicillin-resistant *Staphylococcus aureus* (LA-MRSA) has frequently been reported, posing significant zoonotic concern worldwide. In this study, the occurrence of novel types of SCC*mec* IX elements was identified in two clonal complex (CC) 398 LA-MRSA strains derived from a pig farm and a slaughterhouse in Korea. **Methods**: Whole-genome sequence analysis of the two CC398 MRSA-SCC*mec* IX strains, designated KF1A-1172 and JS1E-122, revealed that these strains are most closely related to previously characterized strains of CC398 LA-MRSA carrying SCC*mec* V isolated from pig farms in Korea. **Results**: Further structural analysis of the SCC*mec* IX in KF1A-1172 and JS1E-122 revealed the presence of multiple *ccr* gene complexes (*ccrA5B3, ccrAB3*, and a truncated *ccrA1*), including *ccrA1B1* genes for SCC*mec* type IX. In addition, the pseudo-SCC (ΨSCC) elements, genes associated with the type 1 restriction–modification (RM) system, and zinc resistance gene *czrC*, were identified in the SCC*mec* IX. **Conclusions**: These findings suggest that the multiple recombination events of elements derived from various SCC*mec* types contributed to the emergence of the novel SCC*mec* IX variant. The identification of these novel SCC*mec* IX types in CC398 LA-MRSA also suggests that epidemiological diversification of SCC*mec* IX in CC398 LA-MRSA is an ongoing event, necessitating continued surveillance on the emergence of novel SCC*mec* variants. This study is the first to report the complete genome sequences of CC398 MRSA carrying SCC*mec* IX in Korea.

## 1. Introduction

Livestock-associated methicillin-resistant *Staphylococcus aureus* (LA-MRSA) has become a major global food safety concern that can be disseminated from various food-producing animals to humans [1,2]. The most common genotype of LA-MRSA in Europe and North America is a clonal complex (CC) 398 LA-MRSA, which primarily colonizes healthy pigs and nearby farm environments [3]. A high prevalence of CC398 LA-MRSA lineages in pig farms has also been reported in many Asian countries [4,5], including South Korea [6,7]. CC398 LA-MRSA isolates derived from pig farms have been found to frequently carry staphylococcal cassette chromosome *mec* type V (SCC*mec* V) for methicillin resistance [6,7]. However, since the first report of SCC*mec* IX in CC 398 MRSA [8], MRSA isolates harboring SCC*mec* IX have been identified in healthcare facilities and pig farms in Thailand [9,10,11,12], indicative of the potential transmission of SCC*mec* IX between humans and animals. In a previous study from our laboratory, the occurrence of sequence type (ST) 398 LA-MRSA and ST541 LA-MRSA isolates carrying SCC*mec* IX was identified in healthy pigs of different age groups [6]. Given the significant role of SCC*mec* elements in the antimicrobial resistance, molecular diagnosis, and evolution of MRSA isolates, comprehensive knowledge of the structural characteristics of novel SCC*mec* IX elements is necessary. In this study, we identified and characterized two CC398 LA-MRSA isolates (one ST541 and one ST398 LA-MRSA) carrying SCC*mec* IX derived from a pig farm and a slaughterhouse, respectively.

## 2. Results

### 2.1. Genomic Characteristics of the Two CC398 LA-MRSA Strains

Genomic analysis of the KF1A-1172 strain revealed a circular chromosome and four plasmids containing 2699 open reading frames (ORFs), 59 tRNA, and 19 rRNA (Appendix A). The JS1E-122 strain had a circular chromosome and two plasmids containing 2721 ORFs, 59 tRNA, and 19 rRNA. Among the annotated ORFs, 787 (29.2%) genes in KF1A-1172 and 811 (29.8%) genes in JS1E-122 were identified as hypothetical proteins.

Functional categorization of the KF1A-1172 and JS1E-122 genomes by COG classification revealed that the most proteins were assigned to four major COG categories: amino acid transport and metabolism (category E), replication, recombination, and repair (category L), transcription (category K), and inorganic ion transport and metabolism (category P). The prediction of genome annotations based on the SEED database also showed that genes involved in amino acids and derivatives comprised the most abundant subsystem in each genome of KF1A-1172 (229 ORFs) and JS1E-122 (234 ORFs) strains, followed by carbohydrate metabolism (173 ORFs) and protein metabolism (156 ORFs) subsystems, with identical ORF counts in both KF1A-1172 and JS1E-122. Distinct subsystems of annotated genomes between KF1A-1172 and JS1E-122 strains indicated that the ORFs associated with the cell wall and capsule (54 ORFs versus 47 ORFs) and DNA metabolism (68 ORFs versus 61 ORFs) were higher in the KF1A-1172 strain, while the number of ORFs related to phages, prophages, and transposable elements (15 ORFs versus 9 ORFs) and membrane transport (41 ORFs versus 30 ORFs) were higher in the JS1E-122 strain.

### 2.2. Profiles of Antimicrobial and Hevy Metal Resistance Genes in the Two CC398 LA-MRSA Strains

Correlating with the antimicrobial resistance phenotypes (Table 1), the two LA-MRSA strains harbored a variety of ARGs in their genomes. As shown in Appendix A, in silico analyses of ARGs revealed that the genomes of the two strains possessed resistance genes for aminoglycosides [*aadD*, *aac*(6′)-*aph*(2′′), *ant*(6)-*la*, *bleO*, and *str*], β-lactams (*mecA* and *blaZ*), macrolides-lincosamides-streptogramin B (MLS_B_) [*erm*(C) and *erm*(T)], lincosamides [*lnu*(B)], lincosamides-pleuromutilins-streptogramin A [*lsa*(E)], phenicols (*fexA*), tetracyclines [*tet*(M) and *tet*(L)], and trimethoprim (*dfrE* and *dfrG*), and carried dual point mutations in *gyrA* (S84A) and *parC* (S80Y) for quinolones. Three plasmids in the KF1A-1172 strain carried multiple ARGs for aminoglycosides [*aadD*, *aac*(6′)-*aph*(2′′), *ant*(6)-*la*, and *bleO*], MLSs [*erm*(C), *lsa*(E), and *lnu*(B)], and trimethoprim (*dfrE*). In contrast, only a single aminoglycoside resistance gene, *str*, was identified in one of the plasmids in JS1E-122. In addition to ARGs, heavy metal resistance genes (*czrC*, *czcD*, and *cadC*) conferring resistance to zinc chloride were also identified in both the KF1A-1172 and JS1E-122 strains. Notably, the JS1E-122 strain carried the *erm*(T) gene, which was frequently associated with CC398 MSSA strains in previous studies [13,14].

### 2.3. Novel Structures of SCCmec IX Elements in the Two CC398 LA-MRSA Strains

The overall structures of SCC*mec* IX elements in the KF1A-1172 and JS1E-122 strains are illustrated in Figure 1. Consistent with the PCR-based SCC*mec* type classification, SCC*mec*Finder analysis of the SCC*mec* IX elements revealed that the two CC398 LA-MRSA strains carried a type C2 *mec* gene complex and a type 1 *ccr* gene (*ccrA1B1*) complex, indicating a prototype of SCC*mec* type IX based on the criteria of the IWG-SCC [17]. Nucleotide sequence analysis of the entire SCC*mec* IX regions revealed that 69 kb and 77 kb sized SCC*mec* IX were integrated into the *orfX* genes of the KF1A-1172 and JS1E-122 strains, respectively. The SCC*mec* IX elements of the KF1A-1172 and JS1E-122 strains were comparted into three (DR1-4) and four parts (DR1-5), respectively, flanked by DR sequences. Segments flanked by the DR2 and DR3 in the KF1A-1172 strain and DR1 and DR2 in the JS1E-122 strain carried identical *ccrA1* and *ccrB1* genes that showed 97% and 93% nucleotide sequence similarities to those of JCSC6690, and 92% and 90% similarities to those of the JCSC6943 strain, respectively. The SCC*mec* IX regions between DR1 and DR2 in the KF1A-1172 strain and between DR3 and DR4 in the JS1E-122 strain harbored pseudo-SCC (ΨSCC) elements identified in a novel SCC*mec* type V variant of the SHP6P021P strain and shared 96% nucleotide sequence similarity. A novel *ccr* gene complex that consisted of *ccrA* and *ccrB3* genes, which showed 99% sequence similarity to that of the SHP6P021P strain, was identified in the ΨSCC regions of the two CC398 LA-MRSA strains. In addition, a novel *ccr* gene complex was found in the segments between DR3 and DR4 in the KF1A-1172 strain and between DR2 and DR3 in the JS1E-122 strain. The *ccr* genes of the KF1A-1172 and JS1E-122 strains shared 87% nucleotide sequence identity to *ccrA5* and 87% nucleotide sequence identity to *ccrB3*. Flanked by DR3, the SCC*mec* IX region of the JS1E-122 strain contained an additional ΨSCC element, known as ΨSCCJCSC6690, which showed 99% sequence similarity to those of the JCSC6690 strain. Moreover, the JS1E-122 strain had a truncated *ccrA1* with 99% nucleotide sequence similarity to that of the ΨSCCJCSC6690 region. Genes associated with the type 1 restriction–modification (RM) system, consisting of *hsdM2*, *hsdS2*, and *hsdR2*, were also identified within the SCC*mec* IX regions of the KF1A-1172 and JS1E-122 strains. Additional type 1 RM genes (*hsdM1*, *hsdS1*, and *hsdR1*) were identified at the upstream region of *ccrA1B1* in KF1A-1172. The multiple sets of heavy metal resistance genes (*cadDX*, *arsRBC*, *arsDARBC* operons, and *copB* genes) contained in the SCC*mec* IX elements of the JCSC6690 and JCSC6943 strains were not found in the KF1A-1172 and JS1E-122 strains. Instead, a cadmium and zinc resistance gene, *czrC,* was identified downstream of the *ccrA5B3* gene complex. Although the sequences containing *mecA* and *czrC* genes found within the SCC*mec* IX elements of KF1A-1172 and JS1E-122 strains shared > 98% of sequence identity to those in the SCC*mec* V of JCSC6944, *ccr* genes in the two CC398 LA-MRSA strains showed < 50% sequence identity to those of the JCSC6944 strain (Figure 1).

### 2.4. Phylogenetic Lineage of CC398-SCCmec IX LA-MRSA Strains

As shown in Figure 2, a phylogenetic tree was generated based on the SNPs of genome sequences derived from 96 different CC398 *S. aureus* strains (56 MRSA and 40 MSSA strains), including KF1A-1172 and JS1E-122 strains. The classification of SCC*mec* elements in the CC398 MRSA strains revealed that the most common SCC*mec* type was SCC*mec* V (n = 42, 75%), followed by SCC*mec* IV (n = 8, 14.3%), SCC*mec* VII (n = 3, 5.4%), SCC*mec* IX (n = 2, 3.6%), and non-typeable (n = 1, 1.7%). KF1A-1172 and JS1E-122 strains were most closely clustered with the CC398 MRSA strains carrying SCC*mec* V isolated in Korea: K12PJN53 (ST541-MRSA-SCC*mec* V-t324-*agr* I), PCFA-221 and PCFH-226 (ST541-MRSA-SCC*mec* V-t034-*agr* I), and PJFA-521 and PJFE-503 strains (ST398-MRSA-SCC*mec* V-t18103-*agr* I), respectively.

## 3. Discussion

Although SCC*mec* V has been predominantly associated with CC398 LA-MRSA isolates [3,6,7], recent studies have highlighted a high degree of sequence rearrangements within SCC*mec* elements [18,19,20,21], resulting in changes in the SCC*mec* types or subtypes. In this study, two CC398 LA-MRSA strains, KF1A-1172 and JS1E-122, carrying SCC*mec* IX were recovered from a pig farm and a slaughterhouse in Korea. The molecular characterization of these two strains revealed that KF1A-1172 belonged to an ST541 lineage with *spa* type t2876 and *agr* type I, whereas JS1E-122 belonged to an ST398 lineage with *spa* type t571 and *agr* type I (Table 1). Although the locations of the ARGs differed between the genomes of the KF1A-1172 and JS1E-122 strains, the two MRSA strains carried a variety of ARGs responsible for multiple antimicrobial resistance (MDR) phenotypes. In line with previous publications [7,22], the KF1A-1172 and JS1E-122 strains carried a variety of genetic determinants associated with resistance to antimicrobial agents and heavy metals, which have been extensively used in pig farms of Korea.

PCR and in silico analyses of the SCC*mec* elements in the KF1A-1172 and JS1E-122 strains revealed that the two strains had a class type C2 *mec* complex and type 1 *ccr* complex, representing SCC*mec* type IX. The first identification of SCC*mec* IX was described in ST398 and ST9 MRSA strains derived from a veterinarian (JCSC6943) [8] and an outpatient (JCSC6690) [10], respectively, in Thailand. The regions containing the class type C2 *mec* complex and type 1 *ccr* complex displayed a 95–98% nucleotide similarity with those in the JCSC6943 and JCSC6690 strains (Figure 1). However, further structural analysis of the SCC*mec* IX elements in KF1A-1172 and JS1E-122 revealed four distinctive composites within the SCC*mec* IX regions, which had not been reported in previously identified SCC*mec* IX elements.

First, multiple *ccr* gene complexes were present in SCC*mec* IX in KF1A-1172 and JS1E-122. In addition to carrying the *ccrA1B1* gene for SCC*mec* type IX, novel *ccr* gene complexes, including *ccrAB3*, *ccrA5B3*, and truncated *ccrA1*, were dispersed across the different segments of SCC*mec* IX. The *ccrAB3* and truncated *ccrA1* genes in the KF1A-1172 and JS1E-122 strains displayed 99% nucleotide sequence similarity to those of the SHP6P021P and JCSC6690 strains, respectively. Notably, a novel *ccr* gene complex assigned to the *ccrA5B3* allotype, which did not correspond to any known combination of *ccr* complex in SCC*mec* type classification, was identified in SCC*mec* IX in KF1A-1172 and JS1E-122. While several SCC*mec* variants with novel *ccr* complexes have recently been reported [19,21,23], the precise roles of multiple Ccr recombinases in the excision and integration of SCC*mec* elements remain unclear.

Second, SCC*mec* IX in the KF1A-1172 and JS1E-122 strains harbored the SCC-like regions designated as ΨSCC elements. According to recommendations of IWG-SCC, ΨSCC elements are defined as SCC-like regions but do not carry *ccr* genes [17,24]. The novel ΨSCC region containing the *ccrAB3* gene and type 1 RM system genes, previously identified in a novel SCC*mec* V element of the ST398 MRSA-t571 strain (SHP6P021P) isolated from a pig farm in China [18], was integrated into SCC*mec* IX of KF1A-1172 and JS1E-122. The sequences of the ΨSCC regions in the two study strains showed >99% nucleotide sequence similarities with those in the SHP6P021P strain. In addition, the SCC*mec* IX element in JS1E-122 carried the ΨSCCJCSC6690 sequences, previously identified in the SCC*mec* IX [10] and SCC*mec* XII [21] elements. Moreover, a novel SCC-like region harboring the novel *ccrA5B3* gene complex and *czrC* gene was also found in KF1A-1172 and JS1E-122. Although the partial sequences from the *czrC* gene to DR3 and DR4 or DR2 and DR3 in KF1A-1172 and JS1E-122 exhibited 99% nucleotide sequence similarities with those of SCC*mec* type V in the JCSC6944 strain, the entire sequences could not be aligned to any of the previously characterized SCC*mec* types.

Third, the presence of multiple sets of type I RM system genes was found within the SCC*mec* IX elements of the CC398 MRSA strains used in this study. The type I RM system plays an important role in the DNA modification and evolution of the staphylococcal genome by influencing horizontal gene transfer, genome stability, and recombination events, such as the integration and reconstruction of SCC*mec* [25,26]. As indicated by previous studies [18,27], SCC*mec* elements harboring multiple type I RM systems represent novel hybrid SCC*mec* variants with diverse genetic compositions originating from other types of SCC*mec* elements. The carriage of multiple type I RM systems may facilitate the modification of DNA at different recognition sites and provide genetic conditions favorable for the modification, recombination, and integration of additional SCC-like elements, resulting in an emergence of novel SCC*mec* variants. Based on the locations of DRs and the structural organization of SCC*mec* IX in KF1A-1172 and JS1E-122, it has been hypothesized that the backbone structure was generated by the integration of the fragments containing the class type C2 *mec* complex, type 1 *ccr* complex, and ΨSCCIX. Subsequently, the ΨSCC segment containing the *hsdM2*, *hsdS2*, *hsdR2*, and *ccrAB3* genes may have been incorporated into the backbone structure, followed by the integration of ΨSCCJCSC6690 sequences into the downstream region of the *ccrAB3* gene complex in the JS1E-122 strain.

Lastly, unlike the SCC*mec* IX elements previously identified in JCSC6690 [10] and JCSC6943 [8], which contained various sets of heavy metal resistance genes, KF1A-1172 and JS1E-122 did not carry any of these resistance genes in their SCC*mec* IX elements. Instead, the *czrC* gene conferring resistance to cadmium and zinc was located downstream of the *mecA* gene complex in KF1A-1172 and JS1E-122. Correlating with the carriage of *czrC*, both KF1A-1172 and JS1E-122 showed zinc chloride resistance phenotypes (Table 1). Several previous studies demonstrated that the localization of *czrC* within the SCC*mec* V element contributes to the predominance of CC398 LA-MRSA-SCC*mec* V in swine farms, where zinc had been used as a dietary supplement [3,7,16,28,29]. Similarly, the localization of *czrC* within SCC*mec* IX in CC398 LA-MRSA may enhance the survival and proliferation of this genetic lineage of strains under zinc selective pressure in pig farms in Korea. The results of the current study suggest that the novel SCC*mec* IX element is distributed among the MRSA strains colonizing pigs, pig farms, and slaughterhouses in Korea. Further research is warranted to investigate whether these novel genetic types of LA-MRSA strains containing SCC*mec* IX can replace the genetic lineages of LA-MRSA carrying SCC*mec* V in pig farms of Korea. Moreover, it is important to understand the intra- and interspecies transmission dynamics of the novel SCC*mec* IX elements.

## 4. Materials and Methods

### 4.1. Bacterial Strains

The two CC398 LA-MRSA strains, KF1A-1172 and JS1E-122, included in this study were each isolated from a pig farm (via nasal swab) in Kyeonggi province and a slaughterhouse (via floor swab) in Jeolla province, Korea, in 2020, respectively [6]. The two MRSA strains were identified using a matrix-associated laser desorption ionization-time of the flight-mass spectrometer (MALDI-TOF MS) (BioMérieux, Marcy L’Etoile, France) and sequencing of 16S rRNA and *tuf* genes (Bionics, Seoul South, Republic of Korea). SCC*mec* typing via multiplex PCR (M-PCR) analysis revealed that both the KF1A-1172 and JS1E-122 strains possessed a combination of *ccr* and *mec* gene complexes (1C2) indicating SCC*mec* type IX for methicillin resistance [30], as prototypic elements of SCC*mec* type IX in the JCSC6943 strain [8]. Further molecular characterizations of the two MRSA strains revealed that KF1A-1172 was ST541-MRSA-SCC*mec* IX-*spa* t2876-*agr* I, and JS1E-122 was ST398-MRSA-SCC*mec* IX-*spa* t571-*agr* I (Table 1).

According to the CLSI guidelines [15], susceptibility to antimicrobials was determined [6]. Both the KF1A-1172 and JS1E-122 strains exhibited resistance phenotypes to ampicillin (AMP), cefoxitin (CEF), chloramphenicol (CHL), ciprofloxacin (CIP), clindamycin (CLI), erythromycin (ERY), fusidic acid (FUS), gentamicin (GEN), and tetracycline (TET). KF1A-1172 showed additional resistance to sulfamethoxazole-trimethoprim (SXT) and quinupristin-dalfopristin (SYN). The minimum inhibitory concentrations (MICs) (256 µg–0.016 µg) to oxacillin (OXA) and tetracycline were determined using a standard E-test (BioMérieux) on Mueller–Hinton agar plates. The MIC to zinc chloride was determined as previously described [7,16]. *S. aureus* ATCC 25923 and *S. aureus* ATCC 29213 were used as quality control strains for antimicrobial and zinc chloride susceptibility assays. The genotypic and phenotypic profiles of the two CC398 MRSA strains are shown in Table 1.

### 4.2. Whole-Genome Sequence Analysis

Genomic DNA (gDNA) samples of the KF1A-1172 and JS1E-122 strains were extracted using a Wizard genomic DNA Isolation Kit (Promega, Madision, WI, USA), according to the manufacturer’s instructions. The quantity and quality of the extracted gDNA were evaluated using a NanoDrop 2000c spectrophotometer (Thermo Scientific, Wilmington, DE, USA).

Whole-genome sequence data of the KF1A-1172 and JS1E-122 strains were generated via a combination of Oxford Nanopore MinION (Oxford Nanopore Technologies, Oxford, UK) and the Illumina iSeq platform (Illumina, San diego, CA, USA). The library preparation of de novo genome assembly was performed by MinION reads in Trycycler v.0.3.0 (https://github.com/rrwick/Trycycler/releases/tag/v0.3.0) (accessed on 27 May 2020). MinION sequencing of KF1A-1172 yielded 637,847 reads covering 2,887,931 bp with a genome coverage of 330x. After de novo assembly, Illumina iSeq, generating 150 bp paired-end reads, was applied for rectifying the error using the Pilon program (version 1.21). Following the hybrid assembly, 3,363,880 reads covering 2,831,312 bp with 167× coverage were generated from Illumina sequencing. The MinION sequencing of JS1E-122 yielded 39,099 reads covering 2,901,962 bp with a genome coverage of 94x. After the de novo assembly, Illumina iSeq, generating 150 bp paired-end reads, was applied for rectifying the error using the Pilon program (version 1.21). Following the hybrid assembly, 1,671,782 reads covering 2,901,962 bp with 76x coverage were generated from Illumina sequencing. Genome annotation was performed using Prokka v1.12b [31]. Cluster of Orthologous Groups (COGs) and the RAST server of SEED databases (http://rast.theseed.org/FIG/rast.cgi (accessed on 27 May 2020)) were used to categorize functional genes in annotated genomes [32].

The presence of antimicrobial and heavy metal resistance genes in the two MRSA genomes was predicted using Resistance Genes Identifier 6.0.3 (RGI) (https://card.mcmaster.ca/analyze/rgi (accessed on 27 May 2020)) based on the Comprehensive Antibiotic Resistance Database (CARD) and Basic Local Alignment Search Tool (BLAST) analysis. The RGI predictions of antimicrobial resistance genes (ARGs) were confirmed with ResFinder 4.6.0 of the Center for Genomic Epidemiology (CGE) (http://www.genomicepidemiology.org (accessed on 27 May 2020)). The SCC*mec* types of the MRSA strains were determined by using SCC*mec*Finder 1.2 (https://cge.food.dtu.dk/services/SCCmecFinder (accessed on 27 May 2020)).

### 4.3. Phylogenetic Analysis

Phylogenetic analysis was performed as previously described [33]. Briefly, the genome sequences of KF1A-1172 and JS1E-122 were compared with a set of 94 CC398 *S. aureus* strains previously identified in 19 different countries (n = 89) [3], including Korea (n = 5) [33,34,35]. The coding sequences of the genomes were predicted and annotated using Prokka, followed by pan-genome analysis using Roary [36]. All core genes were aligned by PRANK within the Roary pipeline, and a maximum-likelihood tree was calculated by RAxML-HPC2 on XSEDE with a GTRGAMMA model through the CIPRES Science Gateway portal (http://www.phylo.org (accessed on 27 May 2020)) [37]. The phylogenetic relatedness was visualized and annotated with the interactive tree of life (iTOL v5) webserver [38].

### 4.4. Structural Analysis of SCCmec IX

The SCC*mec* IX regions, harboring the *mec* and *ccr* gene complexes, in the KF1A-1172 and JS1E-122 strains were determined according to the International Working Group on the Classification of Staphylococcal Cassette Chromosome Elements (IWG-SCC) recommendations [17]. Compartments of SCC*mec* IX were defined based on the integration site sequence (ISS) comprising the 3′ end of open reading frame X (*orfX*) and flanking direct repeat (DR) sequences. Composites of the SCC*mec* IX cassette indicated that the KF1A-1172 and JS1E-122 strains harbored 68 kb (genomic location: 2444588-2513447) and 77 kb (genomic location: 1545837-1623218) of SCC*mec* IX elements, respectively. For a comparative analysis of SCC*mec* IX elements, the nucleotide sequences of SCC*mec* IX in the KF1A-1172 and JS1E-122 strains were aligned with representative SCC*mec* types by using BLAST and Easyfig [39]: ST398 MRSA-SCC*mec* type V strain JCSC6944 (GenBank accession no. AB505629) [8], ST398 MRSA-SCC*mec* type V variant strain SHP6P021P (GenBank accession no. MN220716) [18], ST398 MRSA-SCC*mec* type IX strain JCSC6943 (GenBank accession no. AB505628) [8], and ST9 MRSA-SCC*mec* type IX strain JCSC6690 (GenBank accession no. AB705452-AB705453) [10].

### 4.5. Accession Numbers

The complete genome sequences of KF1A-1172 and JS1E-122 were deposited in GenBank under the accession no. CP085313-CP085317 and CP085310-CP085312, respectively.

## 5. Conclusions

To the best of our knowledge, this study is among the first to report on the complete genome sequences of CC398 LA-MRSA strains carrying novel types of SCC*mec* IX derived from the pork production system in Korea. The complete genome sequence information will contribute to the understanding of the genomic features and evolutionary dynamics of CC398 LA-MRSA carrying SCC*mec* IX. Moreover, the identification of the novel SCC*mec* IX types in CC398 LA-MRSA suggests that the evolution and diversification of SCC*mec* elements in CC398 LA-MRSA are ongoing events, requiring continued surveillance on the emergence of novel variants of SCC*mec* elements.

## Figures and Tables

**Figure 1 antibiotics-14-00217-f001:**
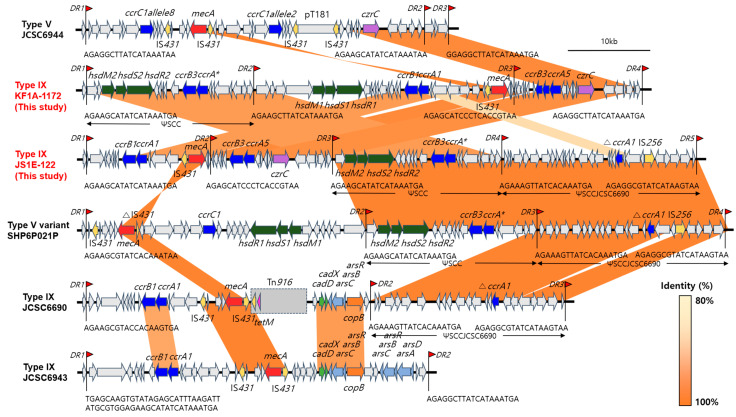
Schematic presentation of SCC*mec* regions in CC398 MRSA strains. All SCC*mec* elements are aligned with *orfX*. Red triangles indicate the location of DRs in SCC*mec* regions with the respective sequences. Different colored arrows indicate different genes with the direction of transcription: red arrows represent *mecA* genes, yellow arrows represent *IS431* or *IS256* elements, blue arrows represent *ccr* genes, a pink arrow represents a *tet*(M) gene, purple arrows represent *czrC* genes, orange arrows represent *copB* genes, light green arrows represent cadmium resistance genes, sky blue arrows represent arsenic resistance genes, and dark green arrows represent *hsd* genes. *ccrA** gene indicates the novel allotype as previously described [18]. Shaded regions indicate nucleotide sequence identities ranging from 80% to 100%.

**Figure 2 antibiotics-14-00217-f002:**
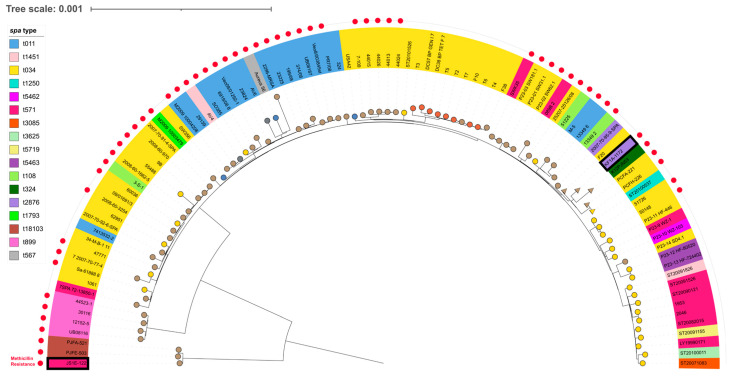
Phylogenetic analysis of CC398 MRSA and MSSA strains. Each circle and triangle in the inner half circle represents an *S. aureus* strain belonging to ST398 or ST541, respectively. The two CC389 MRSA strains from this study are highlighted with light blue boxes. Colored ranges in the left box indicate *spa* types of CC398 *S. aureus* strains, while red circles indicate CC398 MRSA strains.

**Table 1 antibiotics-14-00217-t001:** Genotypic and phenotypic features of the two CC389-MRSA-SCC*mec* IX strains.

Strain	KF1A-1172	JS1E-122
Source	Farm pig	Slaughterhouse environment
MLST-SCC*mec*-*spa*-*agr*	ST541-SCC*mec* IX-t2876-I	ST398-SCC*mec* IX-t571-I
Resistance profiles ^1^	AMP-CEF-CHL-CIP-CLI-ERY- FUS-GEN-SXT-SYN-TET	AMP-CEF-CHL-CIP-CLI-ERY-FUS-GEN-TET
ARG profiles ^2^ (Contig)	Chromosomes: *aac*(6′)-*aph*(2′′), *mecA*, *blaZ*, *erm*(T), *fexA*, and *tet*(M) Plasmids: *aac*(6′)-*aph*(2′′), *ant*(6)-*Ia*, *lsa*(E), and *lnu*(B) (2) *aadD*, *dfr*E, *bleO* (3), and *erm*(C) (5)	Chromosomes: *aac*(6′)-*aph*(2′′), *ant*(6)-*Ia*, *aadD*, *mecA*, *blaZ*, *lsa*(E), *lnu*(B), *erm*(T), *fexA*, *tet*(M), *tet*(L), and *dfrG* Plasmids: str (2)
Virulence gene profiles	*hlb*, *hlgABC*, and *sel26*	*hlb*, *hlgABC*, and *sel26*
OXA MICs (µg/mL) ^3^	128	64
TET MICs (µg/mL) ^3^	64	64
Zinc chloride MICs ^4^	10	10

^1^ AMP, ampicillin; CEF, cefoxitin; CHL, chloramphenicol; CIP, ciprofloxacin; CLI, clindamycin; ERY, erythromycin; FUS, fusidic acid; GEN; gentamicin, SXT, sulfamethoxazole-trimethoprim; SYN, quinupristin/dalfopristin; and TET, tetracycline. ^2^ ARG, antimicrobial resistance gene. ^3^ Minimum inhibitory concentration (MIC) values of 4 ≥ and 16 ≥ µg/mL to oxacillin (OXA) and TET indicate OXA and TET resistance, respectively. The breakpoints of OXA and TET of *S. aureus* are followed according to the CLSI standard [15]. ^4^ MIC values of > 2 mM zinc chloride indicate zinc chloride resistance. The breakpoint of zinc chloride of *S. aureus* is followed according to a previous report [16].

## Data Availability

Data are contained within the article and Appendix A.

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
