# Peer review of "Novel Variants of SCCmec Type IX Identified in Clonal Complex 398 Livestock-Associated Methicillin-Resistant Staphylococcus aureus from Pork Production Systems in Korea"

_antibiotics, 2025, doi:10.3390/antibiotics14030217_

Round 1
Reviewer 1 Report
Comments and Suggestions for Authors
This manuscript describes genomic characteristics of ST398 LA-MRSA in Korea, showing SCCmec IX. This report is a precious and interesting, because CC398 S. aureus is one of the hot topics. in this research field. Although the manuscript is generally well presented, following revision points should considered to improve the quality of this manuscript.
- Authors mentioned "pseudo-SCC" in Results section. (line 112~). It is preferable to add definition and/or evidence by which authors judged the SCC as the pseudo SCC. In Figure 1, schema of KF1A-1172 and JS1E-122 includes ψSCCIX, in addition to other ψSCC. How did authors designate them and are they appropriate descriptions?
- line 152-156 is confusing. Most closely clustered strains of two strains are shown together. Described it by separating the two strains, or use "respectively" appropriately.
- Figure 2, legend. "Each circle and triangle" should be "Each circle and triangle in inner half circle" for better understanding by readers. Present study SCCmec IX strains are highlighted by light blue boxes, but it is somewhat difficult to find. Box with thick black line may be preferable.
- Table 1 is a summary of profiles of the two MRSA strains. But it seems to be too simplified. Though detailed information are shown in Table S2 and S3, some information about the presence or absence of erm(B), erm(C), erm(T), sh-fabI woyld be helpful for understanding the genetic traits of the strains.
- Authors did not mention the presence of virulence factors. Particularly, the presence or absence of ICE genes, chp, scn, sak is notable. Add this information in Table 1 or main text.
- JS1E-122, spa-t571 has erm(T). Such traits has been reported in MSSA, so MRSA may be rare. Please carefully check following articles and discuss the novelty of the Korean CC398 MRSA, with the view from also the presence of erm(T), scn, chi, sak.
(1) Tegegne et al. J Glob Antimicrob Resist. 2022 Jun;29:120-123.
doi: 10.1016/j.jgar.2022.02.017.
(2) Di Gregorio S, et al. Microb Genom. 2023 May;9(5):mgen001020.
doi: 10.1099/mgen.0.001020.
Reviewer 2 Report
Comments and Suggestions for Authors
This is an interesting study with only a few minor issues that need to be adjusted to make a good study even a but better.
Points of criticism:
Lines 55/56 and 57: It would be more informative if the authors could indicate the sizes of the four and two plasmids, respectively.
Line 80/81: I suggest to provide a more detailed description of the MLS resistances, e.g. macrolides-lincosamides-streptogramin B (MLSB) [erm(C) and erm(T)], lincosamides [lnu(B)], as well as lincosamides-pleuromutilins-streptogramin A [lsa(B)]
Lines, 81, 85: trimethoprim
Lines 83-86: It would be interesting to the readers to know which plasmids carry which resistance genes.
Line 92: tetracycline
Line 95: The CLSI document cited is not a guideline, but a standard.
Line 103: A reference for the IWG-SCC is missing.
Lines 138-145: It would be helpful to provide the database accession numbers for all the strains shown in Figure 1 as well as the positions of the SCCmec elements in the respective genomes.
Line 146: The authors switch between the terms “isolates” and “strains”. These are not synonymous terms. The authors should decided for one term and use it consequently throughout their manuscript.
Line 148: How have these 96 strains been selected? Why have MSSA strains been included when the topic was the phylogeny of CC398-SCCmec IX LA-MRSA?
Line 235: Do both czrC-carrying strains also show elevated MICs for cadmium?
Line 253, 254 and elsewhere: bioMérieux – for all companies mentioned, not only the country, but also the city needs to be mentioned.
Lines 261-269: I am not happy with the very superficial description of the antimicrobial susceptibility testing (AST) part. The authors need to mention the method that they have used for AST and according to which performance standard they have conducted AST. Moreover, they need to mention the quality control strains that they have tested side-by-side with their two MRSA strains. They also need to mention the concentration ranges that they have tested for each antimicrobial agent. Finally, they have to mention according to which breakpoints they have classified their strains as resistant, intermediate or susceptible.
Lines 305-306: This strain collection is a very old one (before 2012 and before SCCmec IX was identified) – why did the authors not use a more contemporary strain collection for their phylogenetic analysis.
